# Collateral Effect of the Coronavirus Disease 2019 Pandemic on Emergency Department Visits in Korea

**DOI:** 10.3390/medicina59010090

**Published:** 2022-12-31

**Authors:** Yeon-Joo Cho, In-Hwan Yeo, Dong-Eun Lee, Jong-Kun Kim, Yun-Jeong Kim, Chang-Ho Kim, Jae-Young Choe, Jung-Bae Park, Kang-Suk Seo, Byung-Hyuk Yu, Won-Kee Lee

**Affiliations:** 1Department of Emergency Medicine, School of Medicine, Kyungpook National University, Daegu 41944, Republic of Korea; 2Intensive Care Unit, Kyungpook National University Chilgok Hospital, School of Medicine, Kyungpook National University, Daegu 41404, Republic of Korea; 3Department of Medical Informatics, School of Medicine, Kyungpook National University, Daegu 41944, Republic of Korea

**Keywords:** COVID-19 pandemic, emergency department, facility utilization, case fatality rate

## Abstract

*Background and Objectives*: The ongoing coronavirus disease 2019 (COVID-19) pandemic represents a global public health crisis that has had a serious impact on emergency department (ED) utilization trends. The aim of this study was to investigate the collateral effects of the COVID-19 pandemic on ED utilization trends by patients with mild and severe conditions as well as on 7-day fatality rates. *Materials and Methods*: We analyzed entries in the Korean National Health Insurance claims database between 1 January 2018 and 31 December 2020. Six target patient groups were identified using the main diagnosis codes in the 10th revision of the International Classification of Diseases. Numbers of patients visiting the ED, their age, regional differences, 7-day fatality rate, and rate of emergency procedures were compared between 2018 and 2019 as the control period and 2020, when the COVID-19 pandemic was in full force. *Results*: During the 2020 COVID-19 pandemic, the number of patients who visited the ED with low-acuity diseases and severe acute respiratory infection diseases sharply decreased to −46.22% and −56.05%, respectively. However, the 7-day fatality rate after ED visits for low-acuity diseases and severe acute respiratory infection diseases increased to 0.04% (*p* < 0.01), and 1.65% (*p* < 0.01), respectively, in 2020 compared to that in the control period. *Conclusions*: During the 2020 COVID-19 pandemic, ED utilization impacted and 7-day fatality rate after ED visit increased. Health authorities and health care providers must strive to ensure prompt delivery of optimal care in EDs for patients with severe or serious symptoms and time-dependent diseases, even during the ongoing COVID-19 or potential future pandemics.

## 1. Introduction

The coronavirus disease 2019 (COVID-19) pandemic is an ongoing global health crisis marked by a constellation of effects on emergency department (ED) utilization trends. Since the Korean “index patient” was reported on 20 January 2020, 35% of the Korean population (18.26 million) were confirmed to have COVID-19 and 24,416 died from it as of 17 June 2022 [1,2,3]. While it would be ideal to allow both COVID-19 and non-COVID-19 patients to seek medical care equally, most countries shifted their public health policies to direct medical attention to COVID-19 patients amid limited medical resources in the early period of the pandemic [2,3,4,5]. In Korea, owing to the lack of clear guidelines for ED triage criteria during the early days of the pandemic, a confirmation of a COVID-19 case in the ED forced a hospital to close its ED temporarily to prevent the spread of the infection. This resulted in the temporary suspension of care in many EDs nationwide [6]. Consequently, ED designation for emergency care for patients being transported in an ambulance or waiting at home was delayed, unfortunately resulting in patient death [7].

Studies were conducted on changes in ED utilization during an infectious disease epidemic [8,9]. During the COVID-19 pandemic, ED utilization by pediatric patients, patients with low-acuity conditions, and those with traumatic injuries dropped by 20–63%, whereas that by patients with high-acuity conditions, such as acute myocardial infarction (AMI), stroke, and metabolic emergencies, did not decrease as much [10,11,12,13,14,15,16,17,18]. However, reduced ED visits can result in excess deaths as a result of suspended or delayed treatment of serious or life-threatening diseases other than COVID-19 [19,20]. In addition, multiple COVID-19 waves impacted patient fatality rates, which the government has tried to overcome by social distancing [21]. 

Analyzing changes in ED utilization patterns during the ongoing COVID-19 pandemic as well as identifying the negative collateral effects of COVID-19 on patients with high-acuity conditions and fatality rates are crucial for the management of current and future novel infectious diseases. Therefore, the aim of this study was to investigate the effects of the COVID-19 pandemic on ED utilization patterns and fatality rates in Korea using claims-based National Health Insurance (NHI) data [22].

## 2. Materials and Methods

### 2.1. Study Setting and Databases

In this nationwide retrospective study, we analyzed entries in the NHI claims database between 1 January 2018 and 31 December 2020. The NHI database is a public database, formed by the National Health Insurance Service, which includes the entire population of South Korea (over 50 million) [22]. The database includes age, sex, region, date of admission, date of discharge, admission route, diagnosis codes according to the 10th revision of the International Classification of Diseases (ICD-10), date of death, and details about which medical services were used during hospitalization. While patient records contain one principal, one secondary, and three additional diagnostic codes, we analyzed only the principal code to increase the specificity of our analysis [23].

### 2.2. Measurement of Variables

We analyzed patients who visited the ED during the study period. A patient visiting the ED was defined as a patient with an emergency facility fee listed in their prescription history.

We selected several major diseases and classified them into low-acuity and high-acuity diseases, based on principal diagnostic codes. Low-acuity diseases (LAD) included acute gastroenteritis (A08-A09), upper respiratory infections (J00-J06), and acute otitis media (H65-H67). High-acuity diseases included severe acute respiratory infections (SARI) (J09-J22), AMI (I21-I22), acute hemorrhagic stroke (AHS) (I60-62), acute ischemic stroke (AIS) (I63), and cardiac arrest (CA) (I46). To increase specificity, AMI, AHS, and AIS were included only when first diagnosed in the ED and were excluded when there was a medical history with the same diagnosis since 2002 in the NHI database [23].

If AIS patients visited the hospital within the “golden hour” (4.5 h after onset of a stroke), tPA (tissue plasminogen activator), a thrombolytic therapeutic agent, is administered to those for whom treatment is indicated. Patients who received tPA treatment were defined as those with a tPA prescription code along with an AIS diagnostic code. Among the AMI patients, patients who underwent emergent vascular interventions (VI) treatment were defined as those with coronary angiography, percutaneous transluminal coronary angioplasty, stent insertion, thrombectomy, off-pump coronary artery bypass surgery, and extracorporeal membrane oxygenation prescription code [24]. 

We defined 2020 as the main COVID-19 pandemic period, and 2018 and 2019 as the control period. The outcomes were the monthly number of ED visits, ratio of ED visits in 2020 to the averaged visits from the same month in the control periods, important treatment progress rate after visiting the ED (for patients with AMI or AIS), and monthly numbers of 7-day fatalities from among patients visiting an ED [25,26].

Fatality rate was calculated as follows and presented as percentage (%).

Fatality rate = 100 × number of deaths within 7 days of an ED visit ÷ number of patients who visited an ED.

To assess the degree of change in ED visit, death, and fatality rate, tPA, tPA rate, VI, VI rate, the ratio were calculated as follows and presented as a percentage (%).

ED visit change (%) = 200 × ED visit in 2020/(ED visit in 2018 + ED visit in 2019) − 100.Deaths change (%) = 200 × Number of deaths within 7 days of an ED visit in 2020/(number of deaths within 7 days of an ED visit in 2018 + number of deaths within 7 days of an ED visit in 2019) − 100.Fatality rate change (%) = 100 × Number of deaths within 7 days of an ED visit in 2020/ED visit of 2020 − 100 × (Number of deaths within 7 days of an ED visit in 2018 + Number of deaths within 7 days of an ED visit in 2019)/(ED visit in 2018 + ED visit in 2019).tPA change (%) = 200 × Number of tPA in 2020/(Number of tPA in 2018 + Number of tPA in 2019) − 100.tPA rate change (%) = 100 × Number of tPA in 2020/ED visit in 2020 − 100 × (Number of tPA in 2018 + Number of tPA in 2019)/(ED visit in 2018 + ED visit in 2019).VI change (%) = 200 × Number of VI in 2020/(Number of VI in 2018 + Number of VI in 2019 − 100.VI rate change (%) = 100 × Number of VI in 2020/ED visit in 2020 − 100 × (Number of VI in 2018 + Number of VI in 2019)/(ED visit in 2018 + ED visit in 2019).

### 2.3. Statistical Analysis

Descriptive statistics were used to describe the basic features of the data. Categorical variables were expressed as frequency (percentage). Monthly number and trends of ED visits and 7-day fatality rate were analyzed during the study period for low-acuity and high-acuity diseases. Moreover, we analyzed differences according to pandemic waves. The start point of pandemic wave was defined as the month that include the first day with >100 new cases of COVID-19 per day for 2 consecutive days, and the end point was defined as the month that include last day before decreasing to <100 cases per day for ≥7 days. The first wave occurred between February and April 2020, the second wave occurred between August and September 2020, and the third wave lasted from November to the end of December 2020. We defined January as prepandemic, May, Jun, July, and October as inter-wave (Appendix A) [21,27]. In addition, the monthly number of tPA use for AIS patients and vascular interventions for AMI patients were analyzed. We analyzed differences according to patient age and region. Based on age, we divided the patients into three groups: children (0–17 years), adults (18–75 years), and the elderly (over 75 years). The regions we divided Korea into were Daegu and Gyeongbuk, where the first COVID-19 outbreak in Korea occurred; Seoul, Gyeonggi and Incheon, where the second wave of the epidemic started; and other regions [21,27]. A chi-square test was used to compare categorical variables. Data management and statistical analyses were conducted using SAS version 9.4 (SAS Institute Inc., Cary, NC, USA), and *p* valued < 0.05 were considered statistically significant.

## 3. Results

We analyzed annual trends of patient visits to EDs for LAD, SARI, AHS, AIS, and AMI from 2018 to 2020. The visits for LAD and SARI showed a decreasing trend over 3 years, especially decreasing sharply in 2020. The number of visits for LAD was 196,564, 174,181, and 99,692 and that for SARI was 77,660, 64,279, and 31,193 in 2018, 2019, and 2020, respectively. The visits for AHS, AIS, and AMI increased in 2019 compared to that in 2018, but decreased in 2020. The number of visits for AHS was 4553, 4881, and 4706; that for AIS was 10,357, 11,792, and 11,468; and that for AMI was 5351, 5594, and 5456 in 2018, 2019, and 2020, respectively. The visits for CA increased, from 3291 in 2019 to 3611 in 2019 and 3915 in 2020 (Appendix A). From February 2020, when the local epidemic began, the number of patients with LAD who visited the ED decreased significantly. By March 2020, visits to the ED by patients with LAD had decreased to less than 50% compared with that during the control period, and since then, the rate has hovered around 50%. The number of SARI patients who visited the ED decreased significantly by February 2020 to 59.29% of the number recorded during the control period; since then, the rate remained below 50% and decreased to only 6.46% in December 2020 (Figure 1 and Appendix A). 

Compared with the control period, ED visit change in 2020 was −46.22% for LAD, −56.05% for SARI, −0.23% for AHS, 3.55% for AIS, −0.30% for AMI, and 13.45% for CA. Compared with the control period, 7-day fatality rate in 2020 increased to 0.04% for LAD (*p* < 0.01), 1.65% for SARI (*p* < 0.01), 0.62% for AHS (*p* = 0.25), 0.25% for AIS (*p* = 0.12), 0.23% for AMI (*p* = 0.58), and 0.46% for CA (*p* = 0.50) (Table 1 and Appendix A). According to the analysis of the epidemic wave period, the 7-day fatality rate increased in all disease groups during total epidemic wave periods and inter-wave periods, but only LAD (*p* < 0.01) and SARI (*p* < 0.01) showed statistical significance (Table 2).

The number of patients who were treated for AIS as the main diagnosis in the ED was 10,357 in 2018, 11,792 in 2019, and 11,468 in 2020. Of those patients who visited the ED and were diagnosed with AIS during the study period, 915 were administered tPA in 2018, 1151 in 2019, and 1081 in 2020. Accordingly, the tPA treatment rates were 8.83%, 9.76%, and 9.43%, respectively. Of all patients who visited the ED and were treated for AMI as the main diagnosis, 4589 underwent vascular interventions in 2018, 4822 in 2019, and 4717 in 2020. Thus, the vascular intervention rates were 85.76%, 86.20%, and 86.46%, respectively, but there was no statistical significance (Table 3).

Our age comparison shows that ED visits for LAD and SARI were significantly reduced in those aged under 18 years compared with those in the other age groups (Appendix A).

In the regional comparison, the ED visit rate of LAD groups in the Daegu–Gyeongbuk region in February 2020, when the COVID-19 epidemic started, centered on Daegu and Gyeongbuk region and showed a tendency to decrease more than that in other regions, and in the second epidemic period, centered on Seoul, Gyeonggi and Incheon region and showed a tendency to decrease more than that in other regions (Appendix A).

## 4. Discussion

This study is the first study to analyze changes in ED utilization pattern during the ongoing COVID-19 pandemic among various patients with low-acuity and high-acuity conditions in Korea using claims-based NHI data. There was a sharp decrease in the number of ED visits for low-acuity conditions, while there a was relatively small change in the number of ED visits for acute and time-dependent diseases, such as AMI, AIS, AHS, and CA (Figure 1).

According to Korea’s National Emergency Department Information System annual statistical report, the total volume of ED visits has declined by 26.07% in 2020 (since the outbreak of COVID-19) compared with that in 2018 and 2019, and ED utilization by pediatric patients, patients with mild traumatic injuries, and those with LAD has also decreased markedly [28]. The observed reduction in ED visits in Korea is, however, relatively small compared with the 20–60% decrease in ED volume observed in other countries during the COVID-19 pandemic (Appendix A) [10,11,12,13,14,15,16,17,18].

The global decrease in total ED volume as a result of the COVID-19 pandemic can be attributed to several causes, such as pure lockdown effects, fear and uncertainty about the novel infectious disease, concerns about the possibility of extended wait times and reduced infection due to improved personal hygiene, public health campaigns to discourage people from overburdening the healthcare system, restricted access to emergency medical care owing to overburdened healthcare facilities, care shifts to other venues and administration methods including telemedicine, and misinterpretation of COVID-19 symptoms [3,6]. It is speculated that the fear of contracting the virus at a hospital is the greatest cause of reduced ED visits by non-COVID-19 patients. Reduced ED utilization for relatively mild causes and non-emergency diseases, such as for patients with minor traumatic injuries or respiratory diseases and pediatric patients, can be considered a positive effect of the pandemic since it alleviates crowding of EDs and thus allows the direction of resources to COVID-19 patients and those with severe conditions. However, it can increase fatality in patients who consider themselves non-emergency and have delayed ED visits.

Studies in other countries reported widely varying changes in ED utilization for AMI, stroke, or Emergency Severity Index level 1–2 conditions, at 10–40% [11,12,13,14,15,19]. However, the incidence of high-acuity diseases, such as AMI and stroke, remained largely unchanged during the pandemic, in contrast to that of minor injuries, pediatric conditions, and respiratory diseases; as the former are time-dependent diseases that require prompt ED visits and care, it is crucial to analyze the causes of even small decreases in ED visits for such diseases and develop measures to address them. At the hospital level, extended prehospital time intervals, increased ED length of stay (LOS), delayed procedures or the inability to provide procedures for stroke and AMI, and delayed intensive care unit (ICU) admissions lead to an exacerbation of neglected pathological conditions and healthcare damage [13,19,20]. Furthermore, patients with non-ST segment elevation myocardial infarction or angina have shown a greater decline in ED utilization than those with AMI, due to the relatively less severe symptoms of the former or due to symptoms being mistaken for COVID-19, and this has been reported to result in delayed treatment and consequent exacerbation of such conditions [19,20]. In our study, the numbers of ED visits with AIS and AMI did not change statistically signifantly. In addition, no statistically significant changes were found in tPA or vascular intervention. However, the numbers of ED visits with LAD and SARI decreased significantly, the fatality rate of LAD and SARI increased statistical significantly (Table 2 and Table 3). Fatality rate could be associated with avoidance of or delayed ED visits by patients.

The all-cause fatality recorded in Korea in 2020 is similar to that recorded for the previous 10 years, but there was excess in-hospital mortality among patients who visited EDs [29,30]. Most patients who died from COVID-19 or its complications dies in the ICU or the isolation ward, rather than in the ED. Moreover, according to Korea’s National Emergency Department Information System annual statistical report, the number of deaths in the ED increased by approximately 8%, which is the most concerning collateral effect of the COVID-19 pandemic on EDs [28]. These increase in deaths are confirmed in published data and studies from various countries, including the United States, as per the Centers for Disease Control and Prevention [31,32]. We speculate that the reasons for the elevated ED and 7-day fatality rate compared with a 26.07% decrease in ED volume are increased ED LOS, inability to admit patients into the ICU or ward; delays of planned surgeries, procedures, and chemotherapy; prehospital delays; limited health care staffing; and limited facility resources. The fact that many hospitals voluntarily closed down their EDs upon suspicion or confirmation of a COVID-19 case in the early phase of the COVID-19 pandemic amid a lack of clear guidelines to prevent the spread of the disease, which resulted in suspension of ED care, may also be another cause [4]. There is an imperative need for research on more accurate ED triage criteria for infectious disease-related and non-infectious disease-related ED visits in preparation of potential outbreaks of novel infectious diseases to prevent such ED closures in the future. Moreover, additional research on the causes of increase in deaths in the ED and measures to address them is crucial to reduce collateral damage from infectious-disease epidemics. 

This study has some limitations. First, we used the official NHI data and final principal diagnosis. Moreover, cases with past diagnosis of high-acuity disease were excluded to increase specificity. That could rule out acute relapses of the same disease, leading to an under-extraction of the entire study population. Second, while we report an increase in 7-day fatality rate, the diagnoses, causes, and pre-existing conditions related to these deaths were not analyzed. Therefore, further research is needed to determine the specific causes underlying the increased fatality rate. Further, studies should examine additional factors, such as time of disease onset, time of visit, ED LOS, time from admission to test and procedure, time of ICU admission, and relevance to COVID-19, which may have an impact on the cause of death. Additionally, studies should examine and analyze other high-acuity conditions, such as cancer, metabolic emergencies, and major trauma, to obtain more objective data on increase in deaths. Third, this study is a retrospective study, and the COVID-19 pandemic is ongoing; fluctuations in the actual numbers of patients therefore need to be considered. Finally, changes in the cost of ED visits, an important part of the impact of COVID-19, have not been studied.

## 5. Conclusions

During the COVID-19 pandemic in Korea, ED utilization and 7-day fatality rate after ED visits changed. ED utilization for LAD and SARI sharply decreased, but the 7-day fatality rate increased. Changes in ED utilization could be associated with changes in fatality rate. Based on our findings, we propose the following crucial measures for ensuring appropraite care for patients and lowering fatality. First, health care providers and health authorities must increase public awareness of the importance of seeking immediate ED care for severe symptoms, diseases, and injuries. Second, infection control measures to protect patients and health care providers are essential to prevent patients from avoiding ED visits due to a fear of infection. Finally, in order to prepare for another potential pandemic, further investigations should be conducted to analyze the causes of the increase in the fatality rate after ED visit.

## Figures and Tables

**Figure 1 medicina-59-00090-f001:**
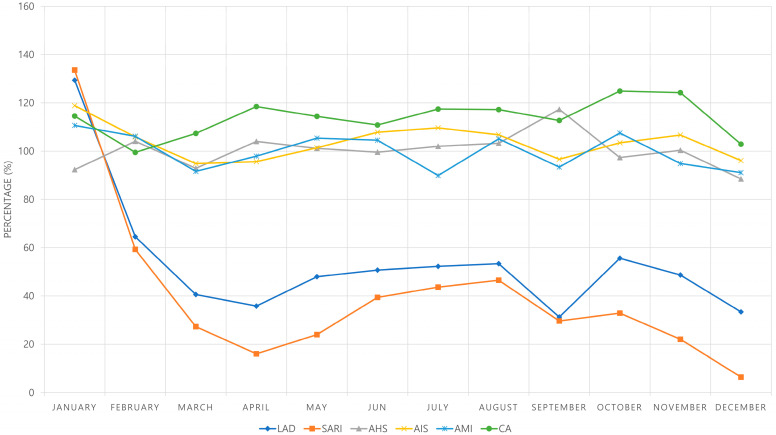
Monthly ED visit rate in 2020 compared to those in the control period. For the investigation of monthly trends from 2018 to 2020, monthly ED visit rates after ED visits for each month in 2020 were compared with the average numbers of the corresponding months in the control period. LAD = Low-acuity disease, SARI = Severe acute respiratory infection, AHS = Acute hemorrhagic stroke, AIS = Acute ischemic stroke, AMI = Acute myocardial infarction, CA = Cardiac arrest, ED = emergency department.

**Table 1 medicina-59-00090-t001:** Deaths within 7 days of an ED visit from 2018 to 2020 listed by diagnosis.

	2018		2019		2020		2020 vs. 2018 & 2019			
	ED Visit, n	Deaths, n (Fatality, %)	ED Visit, n	Deaths, n (Fatality, %)	ED Visit, n	Deaths, n (Fatality, %)	ED Visit Change, %	Death Change, %	Fatality	
									Rate Change, %	*p*-Value
LAD	196,564	69 (0.04)	174,181	58 (0.03)	99,692	71 (0.07)	−46.22	11.81	0.04	<0.01 **
SARI	77,660	979 (1.26)	64,279	931 (1.45)	31,193	935 (3.00)	−56.05	−2.09	1.65	<0.01 **
AHS	4553	466 (10.24)	4881	490 (10.04)	4706	506 (10.75)	−0.23	5.86	0.62	0.25
AIS	10,357	211 (2.04)	11,792	231 (1.96)	11,468	258 (2.25)	3.55	16.74	0.25	0.12
AMI	5351	374 (6.99)	5594	371 (6.63)	5456	384 (7.04)	−0.30	3.09	0.23	0.58
CA	3291	2833 (86.08)	3611	3163 (87.59)	3915	3419 (87.33)	13.45	14.04	0.46	0.50

** *p* < 0.01. ED = emergency department, LAD = low-acuity disease, SARI = severe acute respiratory infection, AHS = acute hemorrhagic stroke, AIS = acute ischemic stroke, AMI = acute myocardial infarction, CA = cardiac arrest.

**Table 2 medicina-59-00090-t002:** Comparison between epidemic wave in COVID-19 pandemic and control periods.

		Prepandemic	1st Wave	2nd Wave	3rd Wave	Total Wave ^a^	Inter-Wave
LAD	ED visit change, %	29.42	−52.44	−58.1	−60.38	−56.47	−48.58
	Deaths change, %	20.00	11.11	−31.43	84.62	4.00	24.32
	Fatality rate change, %	0.00	0.04	0.03	0.08	0.05	0.04
	*p*-Value	0.03 *	<0.01 **	0.14	<0.01**	<0.01 **	<0.01 **
SARI	ED visit change, %	33.61	−66.05	−63.14	−89.67	−77.06	−66.89
	Deaths change, %	−3.64	−11.29	18.03	−2.11	−1.77	−2.03
	Fatality rate change, %	−0.28	2.40	4.47	6.62	4.00	3.87
	*p*-Value	<0.01 **	<0.01 **	<0.01**	<0.01**	<0.01 **	<0.01 **
AHS	ED visit change, %	−7.66	0.08	10.48	−5.81	0.83	−0.07
	Deaths change, %	−30.53	6.56	25.18	−5.62	7.31	14.67
	Fatality rate change, %	−2.73	0.65	1.35	0.02	0.65	1.46
	*p*-Value	0.14	0.54	0.33	0.99	0.36	0.13
AIS	ED visit change, %	18.87	−1.53	1.74	1.33	0.29	5.52
	Deaths change, %	33.33	23.21	2.94	−15.29	5.66	33.33
	Fatality rate change, %	0.25	0.54	0.02	−0.36	0.11	0.49
	*p*-Value	0.66	0.14	0.95	0.36	0.92	0.08
AMI	ED visit change, %	10.69	−1.57	−0.90	−7.06	−3.07	1.77
	Deaths change, %	6.90	11.6	−6.56	−7.44	0.94	5.70
	Fatality rate change, %	−0.22	0.93	−0.39	−0.03	0.28	0.27
	*p*-Value	0.87	0.29	0.70	0.98	0.62	0.71
CA	ED visit change, %	14.56	8.23	15.00	12.48	11.37	17.05
	Deaths change, %	9.61	6.48	19.00	13.12	11.84	19.57
	Fatality rate change, %	−3.84	−1.41	2.96	0.50	0.37	1.86
	*p*-Value	0.08	0.30	0.09	0.74	0.67	0.12

Changes in incidence of ED visits, deaths, and fatality by epidemic wave in 2020 compared to the average of corresponding periods in 2018 and 2019. ^a^ Total wave: February, March, April, August, September, November, and December. * *p* < 0.05, ** *p* < 0.01. ED = emergency department, LAD = low-acuity disease, SARI = severe acute respiratory infection, AHS = acute hemorrhagic stroke, AIS = acute ischemic stroke, AMI = acute myocardial infarction, CA = cardiac arrest.

**Table 3 medicina-59-00090-t003:** Changes in the urgent management for patients with acute ischemic stroke and myocardial infarction in the ED from 2018 to 2020.

			Prepandemic	1st Wave	2nd Wave	3rd Wave	Total Wave ^a^	Inter-Wave	Total
AIS	2018	ED visit, n	815	2408	1700	1853	5961	3581	10,357
		tPA, n (%)	73	205	158	159	522	320	915
		Rate, %	8.96	8.51	9.29	8.58	8.76	8.94	8.83
	2019	ED visit, n	918	2765	1970	2063	6798	4076	11,792
		tPA, n	88	278	192	199	669	394	1151
		Rate, %	9.59	10.05	9.75	9.65	9.84	9.67	9.76
	2020	ED visit	1030	2547	1867	1984	6398	4040	11,468
		tPA	117	250	189	166	605	359	1081
		Rate, %	11.36	9.82	10.12	8.37	9.46	8.89	9.43
	Change	ED visit, %	18.87	−1.53	1.74	1.33	0.29	5.52	3.55
		tPA, %	45.34	3.52	8.00	−7.26	1.60	0.56	4.65
		Rate, %	2.07	0.48	0.59	−0.78	0.12	−0.44	0.10
		*p*-Value	0.12	0.54	0.53	0.37	0.80	0.48	0.79
AMI	2018	ED visit	436	1339	832	932	3103	1812	5351
		VI	365	1150	718	790	2658	1566	4589
		Rate, %	83.72	85.88	86.30	84.76	85.66	86.42	85.76
	2019	ED visit	453	1276	940	1009	3225	1916	5594
		VI	391	1098	803	865	2766	1665	4822
		Rate, %	86.31	86.05	85.43	85.73	85.77	86.90	86.20
	2020	ED visit	492	1287	878	902	3067	1897	5456
		VI	418	1136	772	769	2677	1622	4717
		Rate, %	84.96	88.27	87.93	85.25	87.28	85.50	86.46
	Change	ED visit, %	10.69	−1.57	−0.90	−7.06	−3.07	1.77	−0.30
		VI, %	10.58	1.07	1.51	−7.07	−1.29	0.40	0.24
		Rate, %	−0.08	2.30	2.09	−0.01	1.57	−1.17	0.47
		*p*-Value	0.99	0.60	0.69	1.00	0.57	0.74	0.82

Numbers of patients who were administered tPA among those diagnosed with acute ischemic stroke and numbers of patients who underwent vascular intervention among those diagnosed with acute myocardial infarction in the ED from 2018 to 2020. ^a^ Total wave: February, March, April, August, September, November, and December. ED = emergency department, AIS = acute ischemic stroke, AMI = acute myocardial infarction, tPA = tissue plasminogen activator, VI = vascular intervention.

## Data Availability

Not applicable.

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
