# Peer review of "Collateral Effect of the Coronavirus Disease 2019 Pandemic on Emergency Department Visits in Korea"

_medicina, 2022, doi:10.3390/medicina59010090_

Round 1
Reviewer 1 Report (Previous Reviewer 2)
Comments addressed.
Author Response
Response to Reviewer 1 Comments
There were no comments.
Thanks for good response.

Reviewer 2 Report (New Reviewer)
Dear editor,
Thank you for the kind invitation to review this manuscript. Overall the manuscript is generally well written with some grammatical errors.
Below are my comments for the authors' consideration.
Introduction
- Consider describing the existing literature in different timeframes of the COVID-19 pandemics
-> Why there were periods of reduced utilisations and why there were period of increased utlisation Internationally. What are the literature available for Korea?
- The state of how COVID-19 was managed in Korea is important and should be described for the different waves to provide background information
-> This will help with interpretation of the study results
Methods
- Were there any reason why the NHI records were only evaluated till Dec 2020?
-> The information appears outdated
- Formulas used in the Tables should be described in the methods
-> e.g. ED visit changes, death change and fatality rate change
Results
- Description of the waves of COVID-19 should be described in the methods rather than Table 2
Discussion
- Are there any unique findings noted in this study, relative to what other studies have shown?
-> What are the reasons for these unique findings if any?
- Perhaps an important limitation is how other segments of healthcare utilisation were affected.
-> E.g. primary care, hospitalisation and length of stay, economic cost associated
- Cost of ED visits were not evaluated in this study which is an important limitation
Minor comments
- The authors should consider linking the ED utilization for SARI with high rates of vaccine hesitancy
-> https://pubmed.ncbi.nlm.nih.gov/34452026/
Author Response
Response to Reviewer 2 Comments
Introduction
Point 1>
- Consider describing the existing literature in different timeframes of the COVID-19 pandemics
-> Why there were periods of reduced utilisations and why there were period of increased utlisation Internationally. What are the literature available for Korea?
Response 1> Thanks for your comment.
Base on your comments, I reviewed the introduction again, I found that the ‘Studies have documented increased or decreased ED utilization during an infectious disease epidemic’ sentence was a little different from my intention and needed to be revised.
I revised as ‘Studies were conducted on changes in ED utilization during an infectious disease epidemic’. Especially, most studies related to COVID-19 have concluded that emergency department utilization has decreased. It can be found in reference numbers 10-18.
Thanks again for your good comment.
Point 2>
- The state of how COVID-19 was managed in Korea is important and should be described for the different waves to provide background information
-> This will help with interpretation of the study results
Response 2> Thanks for your comment.
Base on your comments, we've added the following description of the pandemic.
‘In addition, multiple COVID-19 waves impacted patient fatality rates, which the government has tried to overcome by social distancing [21].
Thanks again for your good comment.
Methods
Point 3>
- Were there any reason why the NHI records were only evaluated till Dec 2020?
-> The information appears outdated
Response 3> Thanks for your comment.
Data from the National Health Insurance Service are not collected immediately after the treatment is finished, but only after a certain period of time has elapsed. At the time of the study, data after 2020 were incomplete and were not included.
Thanks again for your good comment.
Point 4>
- Formulas used in the Tables should be described in the methods
-> e.g. ED visit changes, death change and fatality rate change
Response 4> Thanks for your comment.
Base on your comments, table comments (fatality, ED visit change, death change, Fatality rate change, ED visit change, tPA change, tPA rate change, VI change, VI rate change) were moved to measurement of variables.
Thanks again for your good comment.
Results
Point 5>
- Description of the waves of COVID-19 should be described in the methods rather than Table 2
Response 5> Thanks for your comment.
Base on your comments, description of the wave of COVID-19 was moved to method. Thanks again for your good comment.
Discussion
Point 6>
- Are there any unique findings noted in this study, relative to what other studies have shown?
-> What are the reasons for these unique findings if any?
Response 6> Thanks for your comment.
There are several studies on trends in ED utilization impacted by COVID-19, but studies focusing on various disease groups together are rare. The difference from other studies is that it confirmed that different disease groups were affected differently by COVID-19. Thanks again for your good comment.
Point 7>
- Perhaps an important limitation is how other segments of healthcare utilisation were affected.
-> E.g. primary care, hospitalisation and length of stay, economic cost associated
Response 7> Thanks for your comment.
I described as ‘Further, studies should examine additional factors, such as time of disease onset, time of visit, ED LOS, time from admission to test and procedure, time of ICU admission, and relevance to COVID-19, which may have an impact on the cause of death.’ Some of your comments have been included. Regarding economic cost, research couldn’t be conducted in this database. However, study was conducted in relation to the patient's economic status, it was decided to exclude for the study because there was no relevance. Thanks again for your good comment.
Point 8>
- Cost of ED visits were not evaluated in this study which is an important limitation
Response 8> Thanks for your comment.
Base on your comments, I added the sentence that ‘Finally, changes in the cost of ED visits, an important part of the impact of COVID-19, have not been studied.’
Thanks again for your good comment.
Minor comments
Point 9>
- The authors should consider linking the ED utilization for SARI with high rates of vaccine hesitancy
-> https://pubmed.ncbi.nlm.nih.gov/34452026/
Response 9> Thanks for your comment.
Our study is a study on the decrease in utilization on ED due to fear of infection in the early stages of the COVID-19 Pandemic, and the study you mentioned is a study on the hesitancy of the covid-19 vaccine due to the awareness that the risk of COVID-19 infection is low. After discussing, we concluded that it has limitation to connect with our study. Thanks for the good comment, again.

Round 2
Reviewer 2 Report (New Reviewer)
Nil further comments.
English language editing required due to multiple grammatical errors.
This manuscript is a resubmission of an earlier submission. The following is a list of the peer review reports and author responses from that submission.
Round 1
Reviewer 1 Report
The presented manuscript was a study conducted using descriptive analysis without hypothesis tests.
Before concidering publication, there are several points should be properly corrected.
First, not all the high acuity diseases increased prominently fatality chage in 2020 comparing to 2018 & 2019. Acute hemorrhagic strok (AHS), acute myocardial infarction (AMI) and cardiac arrest (CA) were less than 10 % in table 2. and those might be related to nonmeaningful flactuation. Because the month outcomes would be flaten by the year average method. If the authors want to emphasis the impact of epidemic, I suggest the authors to consider to further compare the outcomes in the period of 3 high epidemic waves to those not in the epidemics.
Second, the conclusion in the manuscript did not supported by the study findings. For an examplethe, acute ischemic stroke (AIS) which the fatality was changed most (12.74% in table 2.) showed an incresed ED visit by 3.55% and an increased tissue plasminogen activator use (9.43% in 2020 comparing to the average 9.32% in 2018 & 2019). The above findings did not support the conclusion: "Health authorities and health care providers must strive to ensure prompt delivery of optimal care in EDs for patients with severe or serious symptoms and time-dependent diseases" (line 28-30) and "First, health care providers and health ... Second, ....Finally, ...acute stroke need to be imparted." (line 267-273). It was suggested that the authors should discuss and conclude according to the study findings of each sepecific disease rather than to the assume impressions. Additonally, the concrete measures to underlying causes might not be able to concluded due to the limitation in a descriptive study.
Reviewer 2 Report
The present manuscript presents an analysis of ED utilization in Korea, focusing in severe diseases.
The analysis is sound, however, tables and figures can be improved.
Table 1 is repeated in Figure 1. I suggest to move Table 1 to supplemental material, as lines 113 to 115 include the absolute numbers.
Conversely, Figure 2 is included in Table 2 and I suggest removing Figure 2.
The text in lines 140 to 147 “The number of patients who visited an ED for AHS and died within 7 days was 466 in 2018, 490 in 2019, and 506 in 2020, showing a steady increase over the past 3 years. The number of patients who visited an ED for AIS and died within 7 days was 211 in 2018, 231 in 2019, and 258 in 2020, showing the same steady increase over the past 3 years. The number of patients who visited an ED for AMI and died within 7 days was 374 in 2018, 371 in 2019, and 384 in 2020. The number of patients who visited an ED for CA and died within 7 days was 2833 in 2018, 3611 in 2019, and 3915 in 2020, with the highest number of deaths recorded in 2020 (Table 2 and Table S1).” Is redundant with the table and show quite similar numbers, difficult to assess an ‘steady increase’ with just 3 years. I find more interesting to highlight relative comparisons based on ED visit change and 7-day fatality rate change.
Present table 3 the same as table 2. Not by months. Monthly analysis is awkward and I suggest dividing the year in periods according to events in 2020: January as prepandemic, February-May 1st wave, June-August summer wave, August-October 2nd wave, October-December 3rd wave. And compare these groups of weeks 2020 vs the same weeks of 2018 and 2019, similar to https://www.mdpi.com/2077-0383/11/7/1752.
Lines 178-182: “In the regional comparison, the ED visit rate of LAD groups in the Daegu-Gyeongbuk region in February 2020, when the COVID-19 epidemic started, centered on Daegu and Gyeongbuk region and showed a tendency to decrease more than that in other regions, and in the second epidemic period, centered on Seoul, Gyeonggi and Incheon region and showed a tendency to decrease more than that in other regions (Table S3).” Sure this comparison will benefit from grouping weeks into pandemic periods.
Line 227-229: “The all-cause fatality recorded in Korea in 2020 is similar to that recorded for the previous 10 years, with the number of deaths from COVID-19 at 900 [28,29]. Most patients who died from COVID-19 or its complications dies in the ICU or the isolation ward, rather than in the ED.” Reword, I don’t get the magnitude of 900 covid-19 deaths over the all-cause mortality.
Lines 230-232: “In contrast, the 7-day fatality rate of patients with high-acuity conditions presenting to the ED increased by 0.53%–12.77%, and the number of deaths in the ED increased by approximately 8%, which is the most concerning collateral effect of the COVID-19 pandemic on EDs [27].” This sentence mixes results of the present study with citations, please clarify the origin of the data.
Line 250: “That could make it under-extracted the entire study subjects.” Not clear.
Line 271: Expand “TIA”.